# P-N Junction-Based Si Biochips with Ring Electrodes for Novel Biosensing Applications

**DOI:** 10.3390/bios9040120

**Published:** 2019-10-11

**Authors:** Mahdi Kiani, Nan Du, Manja Vogel, Johannes Raff, Uwe Hübner, Ilona Skorupa, Danilo Bürger, Stefan E. Schulz, Oliver G. Schmidt, Heidemarie Schmidt

**Affiliations:** 1Department Back-End of Line, Fraunhofer Institute for Electronic Nano Systems, Technologie-Campus 3, 09126 Chemnitz, Germany; 2Helmholtz-Zentrum Dresden-Rossendorf, Bautzner Landstraße 400, 01328 Dresden, Germany; 3Leibniz Institute of Photonic Technology, Albert-Einstein-Str. 9, 07745 Jena, Germany; 4Center for Microtechnologies, Chemnitz University of Technology, Reichenhainer Str. 70, 09126 Chemnitz, Germany; 5Institute for Integrative Nanosciences IFW Dresden, Helmholtzstr. 20, 01069 Dresden, Germany

**Keywords:** biochips, impedance spectroscopy, electrical equivalent circuit, biomaterial, *Lysinibacillus sphaericus* JG-A12

## Abstract

In this work, we report on the impedance of p-n junction-based Si biochips with gold ring top electrodes and unstructured platinum bottom electrodes which allows for counting target biomaterial in a liquid-filled ring top electrode region. The systematic experiments on p-n junction-based Si biochips fabricated by two different sets of implantation parameters (i.e. biochips PS5 and BS5) are studied, and the comparable significant change of impedance characteristics in the biochips in dependence on the number of bacteria suspension, i.e., *Lysinibacillus sphaericus* JG-A12, in Deionized water with an optical density at 600 nm from OD_600_ = 4–16 in the electrode ring region is demonstrated. Furthermore, with the help of the newly developed two-phase electrode structure, the modeled capacitance and resistance parameters of the electrical equivalent circuit describing the p-n junction-based biochips depend linearly on the number of bacteria in the ring top electrode region, which successfully proves the potential performance of p-n junction-based Si biochips in observing the bacterial suspension. The proposed p-n junction-based biochips reveal perspective applications in medicine and biology for diagnosis, monitoring, management, and treatment of diseases.

## 1. Introduction

Biochips [1], as one of the most advancing technologies in the biomedical field, have attracted lots of attention in the past decades due to their promising functionalities, e.g., for the detection and recognition of biomaterial in a considerable wide range [2]. In the application field of microbiology, in comparison to optical microscopy, the biochips can prevent human errors and offer faster and easier functional operation as lab measurement tools for biosensing purposes. Thus, the biochips can be helpful for the disease diagnosis with high reliability and time efficiency. Biochips possess many advantages such as mass production, simple immobilization, high density, and high throughput [3].

In this work, the miniaturized p-n junction-based Si biochips are proposed with well-defined gold ring top electrodes and unstructured platinum bottom electrodes, which offer the advantages for sensing the biomaterial such as cost-effectiveness and high portability. The impedance spectroscopy (ImS) [4] has been used to characterize the novel designed biochips. After applying the biomaterial in the Au ring top electrode region, the two-phase electrode structure has been successfully developed and investigated for establishing the functioning electrical equivalent circuit of biochips, which can be utilized for interpreting the impedance properties that recorded between the top and bottom electrodes [5]. Based on the two-phase electrode structure, the straight-forward linear relationship between the specific equivalent circuit parameters and the cell numbers has been discovered, which offers the opportunity for determining biomaterial concentration with low cost and high efficiency [6].

Furthermore, the novel p-n junction-based Si biochips possess the advantages from different perspectives. First, the analysis cost required by p-n junction-based Si biochips for determining the cell density of the biomaterials is considerably lower than that required by other methods [7]. For example, in comparison to analytical techniques such as mass spectrophotometry, gas chromatography, or liquid chromatography [8], the proposed biochips need no special treatments to the biomaterials and they can be kept alive during the detection process. In perspective applications, the number of biomaterials will be determined in a large concentration range by using p-n junction-based Si biochips in conduction with their impedance characterization. Second, the newly developed two-phase electrode structure enhanced the sensitivity of the corresponding equivalent circuit and improved the analytical accuracy of the recorded impedance properties of biochips, which enabled the possibility for sensing biomaterial with considerable low cost [9]. In this paper, the bacteria *Lysinibacillus sphaericus* JG-A12 [10] has been studied due to its potential industrial applications in metal remediation or selective recovery of metals in recycling processes. The S-layer protein in *Lysinibacillus sphaericus* JG-A12 is mainly responsible for such outstanding metal-binding capabilities. The impedance spectroscopy of p-n junction-based Si biochips may offer a new possibility for online monitoring the biomass during the cultivation process.

The paper is structured as follows: In Materials and Methods section, we describe the structure of proposed p-n junction-based Si biochips and introduce electrical equivalent circuit. In Results section, the systematical experimental study of impedance properties of biochips is demonstrated, and the equivalent circuit parameters are extracted. In Discussions section, the origin of two-phase electrode structure is studied and validated by the experimentally recorded impedance data. The paper is summarized and an outlook is given in Conclusion section.

## 2. Materials and Methods

The bacteria used for the investigation in the work is *Lysinibacillus sphaericus* JG-A12, i.e., a Gram-positive, rod-shaped soil bacterium isolated from the uranium mining waste pile "Haberland" near Johanngeorgenstadt in Saxony, Germany. They were cultivated in nutrient broth (8 g/L, Mast Group) overnight in Erlenmeyer flasks at 30 °C with shaking at 100 rpm. Cell density was determined by measuring the optical density at 600 nm (OD_600_) using a UV–Vis spectrometer. Correlation between OD_600_ and cell number was achieved by cell counting under a microscope using a Neubauer counting chamber.

Detection and culture purity of the *Lysinibacillus sphaericus* JG-A12 is usually done with help of microscopy, including the morphology of colonies on agar plates, the growth behavior, even the smell of the culture. To make sure that a microorganism is *Lysinibacillus sphaericus* JG-A12, one has to use genetics means for detecting the 16S rDNA sequence. All these tests are time-consuming. The cell number could be estimated by optical density measurements (OD 600 nm) by putting culture samples into UV–VIS spectrophotometer and using established correlation between cell count by microscopy and OD values. The proposed p-n junction-based Si biochip could be used in a bypass of a culture vessel to measure cell density.

### 2.1. Structural Description 

As illustrated in Figure 1, phosphor or boron ions have been implanted into p- or n-type silicon wafers with a thickness of 525 µm, which results in an n-p junction or a p-n junction, respectively. The 150 nm thick gold (Au) ring top electrodes have been deposited by dc-magnetron sputtering with inner and outer diameters of 6.7 mm and 8.0 mm (Figure 1b). A ring electrode has been chosen because of the homogenous field distribution between top and bottom electrodes. In the work, the biochips PS5 and BS5 have been manufactured, measured, and modeled to analyze influence of bacteria on the biochips. Table 1 lists the overview of the implantation parameters for the manufacturing of the biochips PS5 and BS5 with ring top electrodes.

The impedance characteristics of biochips PS5 and BS5 have been recorded within the frequency range from 40 Hz to 1 MHz under normal daylight at room temperature. These measurements were taken using the Agilent 4294A precision impedance analyzer. In the impedance experiments, the solvent (Deionized water) and the bacteria (*Lysinibacillus sphaericus* JG-A12) are added into the Au ring top electrode region.

In order to visualize the different concentrations of *Lysinibacillus sphaericus* JG-A12, the optical microscopic images have been taken before adding (Figure 2a) and after adding bacteria with corresponding optical density at 600 nm (OD_600_) (Figure 2b–d). The Au ring top electrodes are deposited on a glass slide for utilizing the phase contrast mode of microscope. The OD_600_ is a common measure for microbial cell density, which can be correlated to the cell number per volume depending on the chosen bacteria. In this work, the OD_600_ of 4 up to 16 are applied in the Au ring top electrode region for further impedance characterization, which corresponds to bacteria concentration of 1.23E9 up to 6.15E9 cfu/mL under the assumption that all of the cells are alive.

### 2.2. Modeling

Impedance spectroscopy (ImS) analysis is a well-established method to observe the adhesion of biomaterials because the adhesion changes the electrical behavior of the proposed biochips. Consequently, the first assumption of the electrical equivalent circuit is obtainable based on the electrical properties from the recorded Nyquist plots from biochips [11]. The complex nonlinear least square (CNLS) software is usually used to model and extract the equivalent circuit parameters from the electrical equivalent circuit. As equivalent circuit parameters, resistors or the combination of resistors and capacitors can be used to describe the ohmic or Schottky contacts in the biochips. The parallel RC pair can be used to analyze the full semicircular arc with its center on the real axis in the Nyquist plot [12]. Additionally, ImS on the proposed biochips yields imperfect semicircles with the center below the x-axis and is modeled by using constant phase elements (CPEs) [13]. The importance of CPE was highlighted by Cole and Cole [14] and was considered as an alternating current system response function [15]. CPE admittance is calculated as Y = 1/Z = Q_0_ (jω)^n^, where Q_0_ has the numerical value of admittance at ω = 1 rad/s with the unit S. Thus, the phase angle of the CPE impedance is frequency independent and has a constant value of -(90*n) degrees. By using the CNLS software, the modeling parameters have been iteratively determined. For the CPE component, the parameters RDE (resistance), TDE (relaxation time), and PDE (phase) can be obtained. The resistance part of CPE is determined by RDE, and the capacitive part Cp in CPE can be computed as Cp = (Q_0 *_RDE)^(1/n)^ /RDE, where Ω_max_ is the frequency at which −Im{Z} is maximum on Nyquist plot and Q_0_ is Q_0_ = (TDE) ^(PDE)^/RDE. The electrical properties of the biochips can be derived from the semicircle structure of the impedance spectra in the frequency domain [16]. It has been demonstrated that the experimental impedance characteristics can be modeled by the impedance response of an electrical equivalent circuit, which consists of CPEs, resistors, capacitors, and inductors [17]. The capacitance and resistance are associated with space charge polarization regions and with particular adsorption at the electrode [18], i.e., most of the structures with electrodes normally contain a geometrical capacitance and a bulk resistance in parallel to it [19], which is the same as for the p-n junction-based Si biochips.

In the proposed p-n junction-based Si biochips, the bulk capacitance of the depletion region of the semiconductor and the capacitance of the Schottky contacts between electrodes and semiconductor contribute to the impedance spectra of the biochips. CPEs have been used to model the biochips. The electrical equivalent circuit model of the solo biochips PS5 and BS5 consist of two pairs of CPEs in parallel with resistors (Figure 3a), while the electrical equivalent circuit of the biochips after adding analytes into the Au top electrode region consists of three pairs of CPEs and resistors (Figure 3b). The equivalent circuit parameters Rs and Ls contribute to the lead impedances.

## 3. Results

The impedance characteristics of biochips PS5 and BS5 are studied under the same experimental conditions. During the experiments, firstly, the ImS on solo biochips are measured without adding anything in the ring top electrode region, secondly, the ImS on biochips are recorded after adding 20 µL Deionized (DI) water. In the third step, for both biochips PS5 and BS5, the additional 1–5 µL DI water or bacteria suspension are applied within the ring top electrodes. Each measurement is repeated on individual biochip for 3 times, and the corresponding experimental (circular dots) and modeled results (solid lines) with error bars are shown in Figure 4 and Figure 5 for biochip PS5 and BS5, respectively.

The equivalent experimental results on two PS5 biochips with the same implantation parameters without filling and with 20 µL DI water are demonstrated in Figure 4a and in Figure 4b as black and red dots, respectively, which reveals the reproducible impedance behavior of the PS5 biochips in the frequency domain. In Figure 4a, additional 1–5 µL DI water is applied after adding the 20 µL DI water, while additional 1–5 µL bacteria is applied in Figure 4b. The additionally added DI water results in an increase of corresponding capacitance and resistance parameters in the equivalent circuits in comparison to the ImS from the case after adding 20 µL DI water, whereas the additional 1–5 µL bacteria cause the decrease of both parameters. Moreover, by adding subsequential bacteria volume from 1 µL to 5 µL as shown in Figure 4b, more significant changes in the third semicircle (CPE3 and R3 as illustrated in Figure 3b) can be recorded in comparison to the impedance characteristics in Figure 4a by adding subsequential 1 µL to 5 µL DI water.

Similarly, based on the equivalent experimental results on two BS5 biochips without filling and with 20 µL DI water in the subfigures in Figure 5, the additional 1–5 µL DI water or bacteria are applied after adding the 20 µL DI water in Figure 5a,b, respectively. Note that, for biochip BS5 after adding the additional 1–5 µL DI water (Figure 5a) or 1–5 µL bacteria (Figure 5b), the decrease of corresponding capacitance and resistance parameters (the thinner curves in Figure 5) in the equivalent circuits in comparison to the ImS from the case after adding 20 µL DI water (thicker red curves in Figure 5a,b) can be recorded, which is different with the biochip PS5. Nevertheless, more significant changes from the impedance characteristics of BS5 after adding additional 1 µL–5 µL bacteria (Figure 5b) can be detected than in the case after adding additional 1–5 µL DI water (Figure 5a), which is consistent to the biochip PS5. These results successfully proved that biochips PS5 and BS5 can be used to detect adhesion of *Lysinibacillus sphaericus* JG-A12 in the ring top electrode region.

Note that the resistance of the boron-implanted biochip BS5 (Figure 5) is typically larger than the resistance of the phosphor-implanted biochip PS5 (Figure 4) due to the lower conductivity of the p-type semiconductor in which the holes are majority carriers in comparison to the n-type semiconductor whereas the electrons are the majority carriers. Conductivity is defined by σ = p.e.µh + n.e.µe, where the mobility of holes and electrons are 505 and 1450 cm^2^/Vs, respectively. By adding analytes, the capacitive impedance is decreased and a third semicircle is formed. Moreover, for the BS5 biochip, the dramatic impedance variation is found after adding additional 1 µL of DI water (violet curve in Figure 5a) and 1 µL of bacterial suspension (violet curve in Figure 5b) than that of biochip PS5 (violet curves in Figure 4). The experimental impedance characteristics from the biochips PS5 can be modeled by the equivalent circuit parameters in the equivalent circuit as shown in Figure 3, which consists of two nonideal capacitances or CPEs (Cp1, Cp2), two parallel resistances (Rp1, Rp2), a contact resistance (Rs) and a contact inductance (Ls). ImS modeling of biochip PS5 is shown in Table 2 and Table 3 (first row). 

It should be noted that the equivalent circuit of the impedance spectra of the solo biochip PS5 and biochip PS5 with analytes are different due to the additional appeared semicircle. Based on the experimental impedance characteristics from the biochips PS5 after adding analytes, the composition and cell numbers of analytes added to the Au top ring electrode region for biochip PS5 can be determined by modeling the equivalent circuit parameters in the equivalent circuit as shown in Figure 3, which consists of three nonideal capacitors (Cp1, Cp2, Cp3) and three resistors (Rp1, Rp2, Rp3), a contact resistance (Rs), and a contact inductance (Ls). The corresponding ImS modeling results of the biochip PS5 are shown in Table 2 (with DI water) and in Table 3 (with DI water and bacteria).

Similarly, the ImS characteristics of the biochip BS5 can be modeled with two pairs of resistors and nonideal capacitors (Rp1, Rp2, Cp1, Cp2), contact resistance (Rs), and contact inductance (Ls). Furthermore, for the biochip BS5 the electrical equivalent circuit model for biochip with analytes consists of three pairs of resistance-capacitance in addition to the contact resistance and contact inductance. The corresponding ImS modeling results of the biochip BS5 are shown in Table 4 (with DI water) and in Table 5 (with bacteria).

With the help of optical microscopy, the calibration between the ImS data and cell concentration observed with the optical density at 600 nm (OD_600_) has been considered. Calibration of the biochip is achieved in the volume range from 0 µL to 5 µL bacterial suspension in 20 µL DI water. OD_600_ of 4 corresponds to 2.46E7 cells on the chip if *Lysinibacillus sphaericus* JG-A12 with 1 µL of concentration is applied to 20 µL DI water. For calibration, the dependency of the modeled equivalent circuit elements Rp1, Rp2, Cp1, Cp2 and Rp3 and Cp3 (from impedance modeling) on the nominal number of bacterial cells (from optical microscopy) was evaluated on the basis of the biochip PS5 (Figure 6).

As demonstrated in Figure 6, 4 equivalent circuit parameters Rp1, Cp1, Rp3, and Cp3 from biochip PS5 have been proved to possess the linear dependence with the number of bacteria. Moreover, 3 equivalent circuit parameters Cp1, Rp3, and Cp3 have the linear relationship with the nominal number of bacterial cells for the biochip BS5 in the range from 2.46E7 to 1.23E8 (Figure 7).

Among the modeled electrical elements in the equivalent circuit in Figure 3b, the Rp1 and Cp1 pair represents the Schottky contact at the electrodes/semiconductor interface. If the size of contact area is denoted as A, by adding bacteria suspension to the top electrode region of biochips, the area of the top contact is increased. According to the equation Rp1 = ρ(d/A), where d denotes the thickness of the Schottky barrier, the resistance is reversely related to the area A. Thus, there is reduction in resistance by adding the bacteria suspension. If we consider Cp1 = ε(A/d) with ε as the permittivity of semiconductor, the relationship between Cp1 and A results in the increasing Cp1 with increasing bacteria suspension. The Rp2 and Cp2 pair corresponds to the impedance of semiconductors Si:B in PS5 and Si:P in BS5, and the values are kept the same in Table 2/Table 3 for PS5 and in Table 4/Table 5 for BS5, respectively. The Rp3 and Cp3 pair represents the impedance of bacterial suspension which is added into the Au top electrode region. According to the experimental results from biochip PS5, Rp3 is decreasing linearly with bacteria concentration, while BS5 Rp3 is increasing linearly. Cp3 is increased linearly in both cases.

In detail, the linear impedance change that depends on the bacterial concentration for the Si Biochip PS5 is illustrated in Figure 6 with the 4 model parameters Rp1, Cp1, Rp3, and CP3. Furthermore, the linear impedance change that depends on the bacterial concentration for Si Biochip BS5 is illustrated in Figure 7 with the 3 model parameters Cp1, Rp3, and CP3. Therefore, a multiparameter determination of the bacterial concentration can be performed for both Si Biochips.

## 4. Discussion

Due to the appearance of non-overlapping semicircular curves in the Nyquist plots which have been shown in Figure 4 and Figure 5, it can be directly estimated that the associated resistor (R) and capacitor (C) can be used for describing the physical structures of the biochips [20]. Based on this initial estimation and the experimental impedance data (Figure 8), the transferring of the equivalent circuits, based on the physical mechanism of the biochips (Figure 9a,c) to the ones based on the assumed associated resistor-capacitor RC pairs (Figure 9b,d), can be validated. 

For the proposed p-n junction-based Si biochip, the Schottky contacts are formed at the interfaces both between Au top electrode/semiconductor and between Pt bottom electrode/semiconductor. These two metal/semiconductor Schottky contacts can be represented by one pair of CPE and resistor in the electrical equivalent circuit [21]. As demonstrated in Figure 9a, the p-n junction in the biochip PS5 or BS5 can be described by depletion region capacitor C_dep_ and semiconductor resistor Rss, which can be further converted to the parallel C_2_ // R_2_ pair fashion (Figure 9b), where C_2_ = C_dep_**·**(Q^2^⁄ (1+Q^2^)), R_2_ = R_ss_**·**(1+Q^2^) with the definition of Q = 1/(**ω**·C_dep_·Rss) [22]. Thus, the impedance spectra from solo biochips PS5 and BS5 as depicted in thick black curves in Figure 4 and Figure 5 can be modeled by using equivalent circuits in Figure 9b.

After applying the analytes (DI water or bacteria), a two-phase electrode contact [23] can be developed by using the proposed biochips, where one phase is resulted from the electrodes of biochips and the other phase is related to the added analytes [24]. The Maxwell circuit in Figure 9c can be transferred into Voigt circuit as illustrated in Figure 9d by utilizing the following equations [25]:(1)$$C_{2,3}=2C_{b}{\left(1\mp \;\frac{\frac{Rss}{Rb}-\frac{Cb}{\;Cdep+1}}{k^{1/2}}\right)}^{-1},$$
(2)$$R_{2,3}=\frac{Rb}{2}\left(1\pm \;\frac{\frac{Cb}{Cdep}\;-\;\frac{Rss}{Rb+1}}{k^{1/2}}\right),$$
(3)$$k={\left(\frac{Cb}{Cdep}+\frac{Rss}{Rb}+1\right)}^{2}-\frac{4Cb.Rss}{Cdep.Rb}.$$

The validity of equivalent circuit for biochips after applying bacteria can be described by experimental impedance data as shown in Figure 8. Here, the impedance magnitude of the biochips PS5 and BS5 after adding additional 1–5 µL bacteria has been depended on cell concentration. The overall impedance magnitude at different frequencies is decreasing with increasing bacteria concentration, which indicates the validity of parallel connection of RC pairs for the equivalent circuit of biochips with added bacteria. It means that the RC pair which is corresponding to the added bacteria should be in parallel with the p-n junction-based RC pair as shown in Figure 9d. Thus, it can be concluded that the additional semicircles after adding the bacteria as demonstrated in Figure 4b and Figure 5b are caused by the liquid phase in the two-phase electrode structure.

Based on the development of two-phase electrode contact, the final equivalent circuit for the biochips with analytes can be achieved as shown in Figure 3b, where three RC pairs applied. The two-phase electrode structure offers the opportunity for detecting the adhesion of bacteria by analyzing the impedance changes recorded in the Nyquist plot with higher accuracy and efficiency.

## 5. Conclusions

In this work, we propose the p-n junction-based Si biochips with gold ring top electrodes and unstructured platinum bottom electrodes, which offer a novel possibility for online monitoring the biomass during the cultivation with considerable low cost. The promising reproducible impedance characteristics of two types of p-n junction-based Si biochips, i.e., PS5 and BS5, with different implantation ion types and ion energy but the same ion fluence, are discussed and demonstrate the significant change between the impedance spectra before and after adding the bacteria suspension *Lysinibacillus sphaericus* JG-A12 in the top electrode region. With the help of the developed two-phase electrode structure, the equivalent circuit is designed for the p-n junction-based Si biochips, four modeling parameters for PS5 (Rp1, Cp1, Rp3, and Cp3) and three modeling parameters for BS5 (Cp1, Rp3, and Cp3) reveal the linear dependence relationship with the bacterial concentration. Such linear dependent parameters are useful for the quantitative measure of the bacteria concentration. The limitation of the bacteria detection utilizing p-n junction-based Si biochips strongly depends on inner and outer diameters of the ring top electrode. Thus, for detecting a smaller number of bacteria, the inner and outer diameters should be reduced. In the future, the p-n junction-based Si biochips by adding different live cells or tissues with different cell dimensions will be studied and their further potential in medicine and biology applications will be explored.

## Figures and Tables

**Figure 1 biosensors-09-00120-f001:**
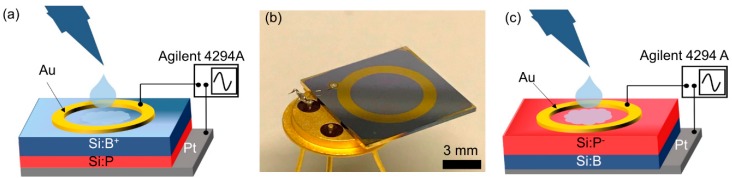
Schematic sketch of the p-n junction-based Si biochip with a ring top electrode with (**a**) boron ions implanted into Si:P or with (**c**) phosphorous ions implanted into Si:B. (**b**) Photograph of a socketed p-n junction-based Si biochip with Au ring electrode. Top and bottom electrodes have been wire-bonded to a diode socket and connected to an Agilent 4294A impedance bridge.

**Figure 2 biosensors-09-00120-f002:**
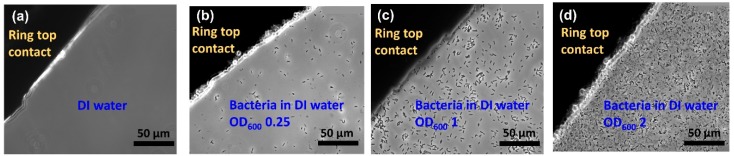
Top view optical microscopic image of a section of ring electrode on glass with (**a**) Deionized water, (**b**) bacteria in water at OD_600_ 0.25, (**c**) bacteria in water at OD_600_ 1, (**d**) bacteria in water at OD_600_ 2 in the ring top electrode region. Here a transparent glass substrate has been used to illuminate the sample with light from the backside. The thickness of the ring top electrodes was 150 nm and was large enough to keep the inserted liquid in the ring top electrode.

**Figure 3 biosensors-09-00120-f003:**
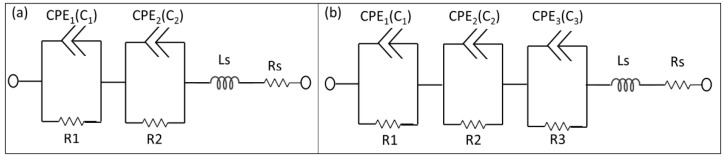
Electrical equivalent circuits used to model impedance spectra of the biochips PS5 and BS5 (**a**) before (two pairs of CPEs and resistors) and (**b**) after inserting analytes into the ring electrode (three pairs of CPEs and resistor). The equivalent circuit parameters Ls and Rs represent the interface properties of the circuit wiring.

**Figure 4 biosensors-09-00120-f004:**
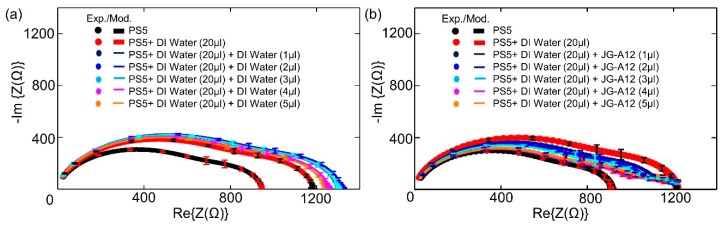
Experimental and modeled Nyquist plots of the Biochip PS5 with (a, b) no filling and with DI water (20 µL) and (**a**) with additional DI water with volume from 1 μL to 5 μL, and (**b**) with additional bacteria volume of JG-A12 from 1 μL to 5 μL. The error bars are inserted in all Nyquist plots according to 3 repeated Biochip experiments. The experimental results are represented in dots, and modeled results are represented in solid lines.

**Figure 5 biosensors-09-00120-f005:**
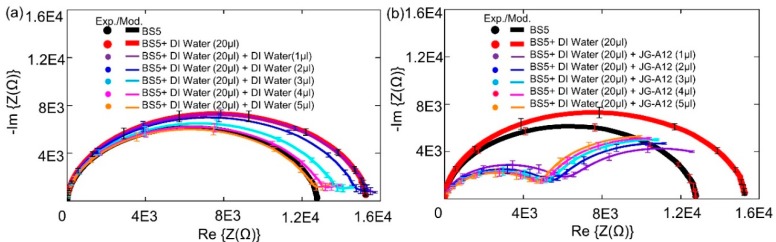
Experimental and modeled Nyquist plots of the Biochip BS5 with (a, b) no filling and with DI water (20 µL) and (**a**) with additional DI water volume from 1 µL to 5 µL and (**b**) with additional bacteria volume of JG-A12 from 1 µL to 5 µL. The error bars are inserted in all Nyquist plots according to 3 repeated Biochip experiments. The experimental results are represented in dots, and modeled results are represented in solid lines.

**Figure 6 biosensors-09-00120-f006:**
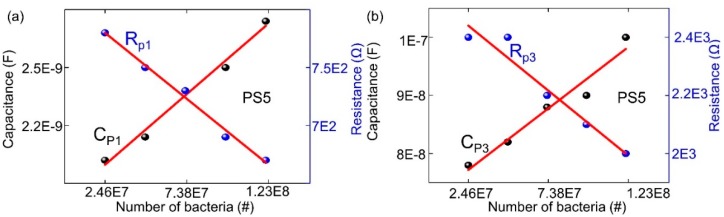
Modeled equivalent circuit parameters (dots) and linear fitting curve (red lines) for (**a**) Rp1, and Cp1 and for (**b**) Rp3 and CP3 of the biochip PS5 in dependence on the number of *Lysinibacillus sphaericus* JG-A12.

**Figure 7 biosensors-09-00120-f007:**
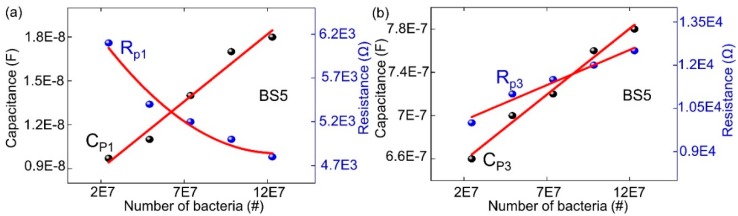
Interpolation (red line) of modeled equivalent circuit parameters (**a**) Rp1 and Cp1 and of (**b**) Rp3 and CP3 of the Si Biochip BS5 that depend on number of *Lysinibacillus sphaericus* JG-A12.

**Figure 8 biosensors-09-00120-f008:**
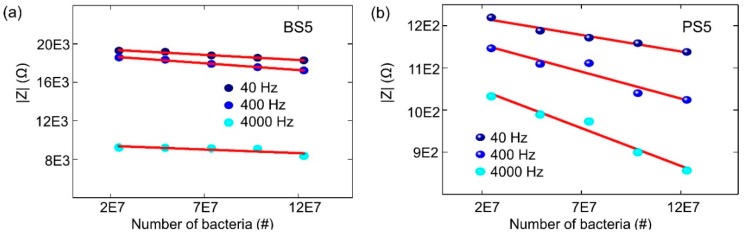
Impedance magnitude measured at test frequencies of 40 Hz, 400 Hz, and 4000 Hz on biochips (**a**) BS5 (Appendix A) and (**b**) PS5 (Appendix A) that depend on the number of bacteria. The experiments are carried out under normal illumination.

**Figure 9 biosensors-09-00120-f009:**
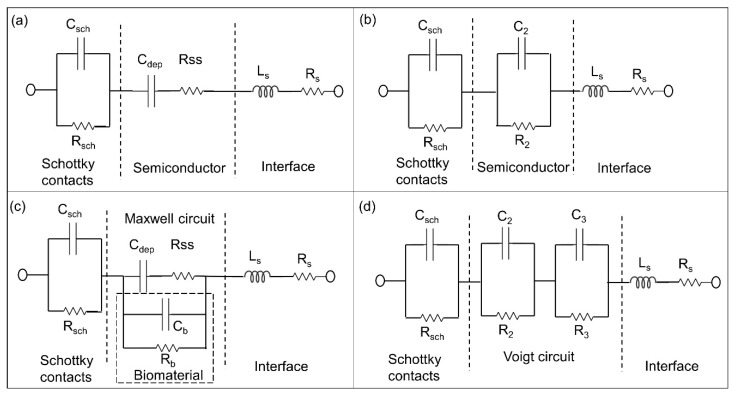
Equivalent circuit models of biochips without analyte bacteria (**a**) based on the physical structure of biochip and (**b**) based on the associated RC pairs. The parallel capacitor C_2_ and resistor R_2_ are transferred from C_dep_ and Rss. Equivalent circuit models of biochip with analyte (**c**) in Maxwell fashion and in (**d**) Voigt fashion. The Voigt fashion can be transferred into Maxwell fashion by using Equations (1)–(3).

**Table 1 biosensors-09-00120-t001:** Implantation parameters of biochips PS5 (phosphor into Si:B) and BS5 (boron into Si:P). The Au ring top electrodes and unstructured Pt bottom contacts have been prepared after ion implantation.

Biochip	Implanted Ion	Ion Energy (MeV)	Ion Fluence (cm^−2^)
PS5	Phosphor	1	3 × 10^13^
BS5	Boron	0.45	3 × 10^13^

**Table 2 biosensors-09-00120-t002:** Modeled equivalent circuit parameters Cp1, Rp1, Cp2, Rp2, Cp3, and Rp3 of the biochip PS5 with 20 µL DI water and different additionally inserted 1–5 µL DI water (Figure 4a).

Circuit Element	Cp1 (F)	Rp1 (Ω)	Cp2 (F)	Rp2 (Ω)	Cp3 (F)	Rp3 (Ω)	Rs (Ω)	Ls (H)
PS5	8.5 × 10^−9^	6.5 × 10^2^	1.4 × 10^−15^	2.8 × 10^6^	-	-	3.6 × 10^−8^	2.4 × 10^−15^
PS5+W20	8.1 × 10^−9^	8.1 × 10^2^	5.1 × 10^−15^	9.4 × 10^6^	1.4 × 10^−7^	2.8 × 10^2^	3.6 × 10^−8^	2.4 × 10^−15^
PS5+W20+W1	7.8 × 10^−9^	8.7 × 10^2^	5.1 × 10^−15^	9.4 × 10^6^	1.4 × 10^−7^	3.7 × 10^2^	3.6 × 10^−8^	2.4 × 10^−15^
PS5+W20+W2	7.6 × 10^−9^	8.7 × 10^2^	5.1 × 10^−15^	9.4 × 10^6^	1.5 × 10^−7^	4.2 × 10^2^	3.6 × 10^−8^	2.4 × 10^−15^
PS5+W20+W3	7.6 × 10^−9^	8.6 × 10^2^	5.1 × 10^−15^	9.4 × 10^6^	1.6 × 10^−7^	4.4 × 10^2^	3.6 × 10^−8^	2.4 × 10^−15^
PS5+W20+W4	7.6 × 10^−9^	8.3 × 10^2^	5.1 × 10^−15^	9.4 × 10^6^	1.6 × 10^−7^	4.4 × 10^2^	3.6 × 10^−8^	2.4 × 10^−15^
PS5+W20+W5	7.6 × 10^−9^	8.1 × 10^2^	5.1 × 10^−15^	9.4 × 10^6^	1.7 × 10^−7^	4.2 × 10^2^	3.6 × 10^−8^	2.4 × 10^−15^

W20 = 20 µL DI water, W1 = 1 µL DI Water, W2 = 2 µL DI Water, W3 = 3 µL DI Water, W4 = 4 µL DI Water, W5 = 5 µL DI Water.

**Table 3 biosensors-09-00120-t003:** Modeled equivalent circuit parameters Cp1, Rp1, Cp2, Rp2, Cp3, and Rp3 of the biochip PS5 and with 20 µL DI water and additionally inserted 1–5 µL bacteria (Figure 4b).

Circuit Element	Cp1 (F)	Rp1 (Ω)	Cp2 (F)	Rp2 (Ω)	Cp3 (F)	Rp3 (Ω)	Rs (Ω)	Ls (H)
PS5	8.5 × 10^−9^	6.5 × 10^2^	5.1 × 10^−15^	9.4 × 10^6^	-	-	3.6 × 10^−8^	2.4 × 10^−15^
PS5+W20	8.1 × 10^−9^	8.1 × 10^2^	5.1 × 10^−15^	9.4 × 10^6^	1.4 × 10^−7^	2.8 × 10^3^	3.6 × 10^−8^	2.4 × 10^−15^
PS5+W20+B1	2.1 × 10^−9^	7.8 × 10^2^	5.1 × 10^−15^	9.4 × 10^6^	7.8 × 10^−8^	2.4 × 10^3^	3.6 × 10^−8^	2.4 × 10^−15^
PS5+W20+B2	2.2 × 10^−9^	7.5 × 10^2^	5.1 × 10^−15^	9.4 × 10^6^	8.2 × 10^−8^	2.4 × 10^3^	3.6 × 10^−8^	2.4 × 10^−15^
PS5+W20+B3	2.4 × 10^−9^	7.3 × 10^2^	5.1 × 10^−15^	9.4 × 10^6^	8.8 × 10^−8^	2.2 × 10^3^	3.6 × 10^−8^	2.4 × 10^−15^
PS5+W20+B4	2.5 × 10^−9^	6.9 × 10^2^	5.1 × 10^−15^	9.4 × 10^6^	9.0 × 10^−8^	2.1 × 10^3^	3.6 × 10^−8^	2.4 × 10^−15^
PS5+W20+B5	2.7 × 10^−9^	6.7 × 10^2^	5.1 × 10^−15^	9.4 × 10^6^	10 × 10^−8^	2.1 × 10^3^	3.6 × 10^−8^	2.4 × 10^−15^

W20 = 20 µL DI water, B1 = 1 µL bacteria, B2 = 2 µL bacteria, B3 = 3 µL bacteria, B4 = 4 µL bacteria, B5 = 5 µL bacteria.

**Table 4 biosensors-09-00120-t004:** Modeled equivalent circuit parameters Cp1, Rp1, Cp2, Rp2, Cp3, and Rp3 of the biochip BS5 with 20 µL DI water and additionally inserted 1–5 µL DI water (Figure 5a).

Circuit Element	Cp1 (F)	Rp1 (Ω)	Cp2 (F)	Rp2 (Ω)	Cp3 (F)	Rp3 (Ω)	Rs (Ω)	Ls (H)
BS-5	4.6 × 10^−9^	1.2 × 10^4^	7.5 × 10^−15^	9.4 × 10^6^	-	-	2.1 × 10^−9^	2.4 × 10^−15^
BS5+W20	4.6 × 10^−9^	1.5 × 10^4^	7.5 × 10^−15^	9.4 × 10^6^	2.6 × 10^−6^	6.2 × 10^5^	2.1 × 10^−9^	2.4 × 10^−15^
BS5+W20+W1	4.5 × 10^−9^	1.5 × 10^4^	7.5 × 10^−15^	9.4 × 10^6^	2.9 × 10^−5^	2.1 × 10^3^	2.1 × 10^−9^	2.4 × 10^−15^
BS5+W20+W2	4.7 × 10^−9^	1.4 × 10^4^	7.5 × 10^−15^	9.4 × 10^6^	3.5 × 10^−5^	2.3 × 10^3^	2.1 × 10^−9^	2.4 × 10^−15^
BS5+W20+W3	4.8 × 10^−9^	1.3 × 10^4^	7.5 × 10^−15^	9.4 × 10^6^	3.9 × 10^−5^	2.5 × 10^3^	2.1 × 10^−9^	2.4 × 10^−15^
BS5+W20+W4	4.8 × 10^−9^	1.2 × 10^4^	7.5 × 10^−15^	9.4 × 10^6^	4.2 × 10^−5^	4.3 × 10^3^	2.1 × 10^−9^	2.4 × 10^−15^
BS5+W20+W5	4.9 × 10^−9^	1.2 × 10^4^	7.5 × 10^−15^	9.4 × 10^6^	4.5 × 10^−5^	5.0 × 10^3^	2.1 × 10^−9^	2.4 × 10^−15^

W20 = 20 µL DI water, W1 = 1 µL DI water, W2 = 2 µL DI water, W3 = 3 µL DI water, W4 = 4 µL DI water, W5 = 5 µL DI water.

**Table 5 biosensors-09-00120-t005:** Modeled equivalent circuit parameters Cp1, Rp1, Cp2, Rp2, Cp3, and Rp3 of the biochip BS5 and with 20 µL DI water and additionally inserted 1–5 µL bacteria (Figure 5b).

Circuit Element	Cp1 (F)	Rp1 (Ω)	Cp2 (F)	Rp2 (Ω)	Cp3 (F)	Rp3 (Ω)	Rs (Ω)	Ls (H)
BS5	4.6 × 10^−9^	1.2 × 10^4^	7.5 × 10^−15^	9.4 × 10^6^	-	-	2.1 × 10^−9^	2.4 × 10^−15^
BS5+W20	4.6 × 10^−9^	1.5 × 10^4^	7.5 × 10^−15^	9.4 × 10^6^	2.6 × 10^−6^	6.2 × 10^5^	2.1 × 10^−9^	2.4 × 10^−15^
BS5+W20 +B1	9.7 × 10^−9^	6.1 × 10^3^	7.5 × 10^−15^	9.4 × 10^6^	5.6 × 10^−7^	1.0 × 10^4^	2.1 × 10^−9^	2.4 × 10^−15^
BS5+W20 +B2	1.1 × 10^−8^	5.4 × 10^3^	7.5 × 10^−15^	9.4 × 10^6^	7.0 × 10^−7^	1.1 × 10^4^	2.1 × 10^−9^	2.4 × 10^−15^
BS5+W20 +B3	1.4 × 10^−8^	5.2 × 10^3^	7.5 × 10^−15^	9.4 × 10^6^	7.2 × 10^−7^	1.1 × 10^4^	2.1 × 10^−9^	2.4 × 10^−15^
BS5+W20 +B4	1.7 × 10^−8^	5.0 × 10^3^	7.5 × 10^−15^	9.4 × 10^6^	7.6 × 10^−7^	1.2 × 10^4^	2.1 × 10^−9^	2.4× 10^−15^
BS5+W20 +B5	1.8 × 10^−8^	4.8 × 10^3^	7.5 × 10^−15^	9.4 × 10^6^	7.8 × 10^−7^	1.2 × 10^4^	2.1 × 10^−9^	2.4× 10^−15^

W20 = 20 µL DI water, B1 = 1 µL bacteria, B2 = 2 µL bacteria, B3 = 3 µL bacteria, B4 = 4 µL bacteria, B5 = 5 µL bacteria

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
