# Peer review of "P-N Junction-Based Si Biochips with Ring Electrodes for Novel Biosensing Applications"

_biosensors, 2019, doi:10.3390/bios9040120_

Round 1

Reviewer 1 Report

The difference between figure 4a and b need be described clearly. How many times of these experiments repeated? error bar should be added in the calibration curves. The English of the whole manuscript should be reorganized.

Author Response

Dear Reviewer, 

thank you for your comments and suggestions! We have prepared our point-by-point response to your comments in the attached PDF file. Please refer to the attachment. 

Thank you in advance! 

Best regards,

Mahdi Kiani on behalf of Heidemarie Schmidt and Nan Du

Reviewer 2 Report

Manuscript Number: biosensors-593867

Title: P-N junction-based Si biochips with ring electrodes 3 for novel biosensing applications

Comments

Authors have demonstrated the p-n junction-based Si biochips with gold ring top electrodes and platinum bottom electrodes and measured impedance in dependence on the number of bacteria such as Lysinibacillus sphaericus JG-A12 in the electrode ring region. An electrical equivalent circuit model is demonstrated to calculate the impedance of the p-n- junction-based biochips and relate the model parameters with the number of bacteria in the electrode ring area. This work is technically to sound to publish in this journal with a revision. My specific comments are given below

The abstract should be re-write with more results. Why this method of detection can provide better performances compared to mass spectrophotometry, gas or liquid chromatography and other methods? Why the Lysinibacillus sphaericus JG-A12 is important to detect? In Fig. 4, the signals (charge transfer resistances in the equivalent circuit) with and without bacteria are not changing so much. Can you authors provide draw a graph to produce linear relation between Rp and bacterial concentration to evaluate the sensing performance? In table 3, why Rp1 values decrease with bacterial concentration? For each measurement, how authors wash the bacterial samples? Authors may need to compare the detection performance with other sensors for detecting this bacteria.

Author Response

(The authors gave the same response as above.)

Reviewer 3 Report

The authors developed a silicon-based biochip for the detection of bacteria adsorption to the sensor surface. The sensing is performed by using a ring shape gold electrode deposited on either a p- or n-type conducting silicon wafer and has been shown to be able to detect JG-A12 adsorption. However, serious concerns are present in the actual form of the manuscript:

1) In the abstract, the authors state that biochips are an alternative to optical microscopy for exploring adsorption. However, biochips were conceived to function inside a living organism, which is quite different to what optical microscopy was conceived for. I do not think the two can be compared.

2) The authors propose a ring design without stating why this is important/the reason behind it.

3) The limit of detection of the device (i.e. minimum number of bacteria to be detected) must be explored and quantified.

4) No error bars/reproducibility estimation is presented.

5) Are the OD_600 nm comparable to clinically relevant concentrations? In terms of cfu/mL, what does this translate in?

6) In the conclusions, it is stated that the design has been optimised but there is no mention against what/how.

*General comment: many of the graphs/tables could be included in the supplementary information rather than in the main body .

Author Response

(The authors gave the same response as above.)

Round 2

Reviewer 2 Report

The authors have improved as the reviewers raised the questions. With this improved version of the manuscript, I would like to recommend to publish this manuscript in biosensors.  

Reviewer 3 Report

Thank you for your edits and efforts in improving the paper and addressing my comments. I have only two further comments/suggestion:s

-The data in tables 2 to 5 could perhaps be easier to understand for a reader if it is plotted as a table.

-On figures 4b and 5b, I would change the label 'bacteria' to 'JG-A12'